# Effect of Cinnamon Extraction Oil (CEO) for Algae Biofilm Shelf-Life Prolongation

**DOI:** 10.3390/polym11010004

**Published:** 2018-12-20

**Authors:** Maizatulnisa Othman, Haziq Rashid, Nur Ayuni Jamal, Sharifah Imihezri Syed Shaharuddin, Sarina Sulaiman, H. Saffiyah Hairil, Khalisanni Khalid, Mohd Nazarudin Zakaria

**Affiliations:** 1Department of Manufacturing and Materials Engineering, Faculty of Engineering, International Islamic University Malaysia, Gombak 50728, Selangor, Malaysia; Mrhaziqhanif@gmail.com (H.R.); ayuni_jamal@iium.edu.my (N.A.J.); shaimihezri@iium.edu.my (S.I.S.S.); sarina@iium.edu.my (S.S.); 2PERMATApintar College, National University Malaysia, Bangi 43600, Selangor, Malaysia; Hanasaffiyah2006@gmail.com; 3Agri-Nanotechnology Program, Biotechnology and Nanotechnology Research Center, Malaysian Agricultural Research and Development Institute (Mardi), MARDI Headquarters, Persiaran MARDI-UPM, Serdang 43400, Selangor, Malaysia; typhloids@gmail.com; 4Department of Biocomposite Technology, Faculty of Applied Sciences, Universiti Teknologi MARA, Shah Alam 40450, Selangor, Malaysia; nazarudin@salam.uitm.edu.my

**Keywords:** cinnamon extraction oil, algae, biodegradation, shelf-life, food packaging

## Abstract

This study was conducted to improve the life-span of the biofilm produced from algae by evaluating the decomposition rate with the effect of cinnamon extraction oil (CEO). The biofilm was fabricated using the solution casting technique. The soil burying analysis demonstrated low moisture absorption of the biofilm, thus decelerating the degradation due to low swelling rate and micro-organism activity, prolonging the shelf-life of the biofilm. Hence, the addition of CEO also affects the strength properties of the biofilm. The maximum tensile strength was achieved with the addition of 5% CEO, which indicated a good intermolecular interaction between the biopolymer (algae) and cinnamon molecules. The tensile strength, which was measured at 4.80 MPa, correlated with the morphological structure. The latter was performed using SEM, where the surface showed the absence of a separating phase between the biofilm and cinnamon blend. This was evidenced by FTIR analysis, which confirmed the occurrence of no chemical reaction between the biofilm and CEO during processing. The prolongation shelf-life rate of biofilm with good tensile properties are achievable with the addition of 5% of CEO.

## 1. Introduction

Quotidian plastic materials for food packaging in the market is proffered using a synthetic polymer base like polyproline and polyethylene, which is strenuous to decompose. Based on every day activities of Malaysian households and industries, high amounts of solid waste materials seem to have polluted and harmed the landfill with poison. In the interest of conserving the landfill, while overcoming the solid waste pollution issues, a few groups of individuals. such as researchers, the government, and industrial players [1], came up with several solutions. Starting off, the government promotes the routine of recycling, reducing, and reusing on a daily basis to the community [2]. Industrial players and researchers further focus on recycling plastics to be used as secondary materials and attempts to recycle and convert those materials into synthetic fiber threads and yarns to produce jerseys, shoes, and other textile products [3]. Nonetheless, the main problems are nowhere near solved at this point. The solid waste keeps on continuing its revolution in our daily life. The development of the first environmentally friendly materials consist partly of conventional plastic polyethylene (PE) or polypropylene (PP) and partly of nature’s own material—chalk (40% by weight). Until early 2004, the research growth on producing the bio-based product was expanding throughout the entire globe [4]. PLA (polylactic acid) films, produced from lactic acid, have shown the highest commercial potential and are now produced on a comparatively large scale. Most bioplastics are produced based on natural biomaterials, such as corn, starch, and soybean [5]. Moreover, other food harvests, for example, cassava, wheat, potato, and sago, have also been transformed into plastic to supplant oil-based plastic [6]. Since those products are food assets for human beings, continuous transformation of those yields into plastic will soon interfere with human sustenance supply by reducing the world’s sustenance assets. In an effort to prevent interference with food assets, other biomaterials should be assessed. Contemporaneous edible films have the potential to substantially reduce the environmental burden due to food packaging and limit moisturization, aroma, and lipid migration between food components [7]. Dealing with the biopolymer materials, it is difficult to avoid fugacious shelf-life issues. Biopolymer, which derives from natural sources, is commonly known for having transitory shelf-life as compared to synthetic biopolymer. Problems and errors may arise during the storage time either for logistic purposes or during transportation of the biopolymer to the industrial consumer. Humidity, temperature, micro-organism, and fungi attack may affect the quality and deteriorate the strength of the biopolymer. Accordingly, this study focuses on producing biofilm with sustainable shelf-life, high quality, low toxicity, and cheap costs, which could provide efficient food chain supply. Therefore, algae are chosen to be used in biopolymer film fabrication. Algae, or seaweed, is an environmental asset that exists in boundless amounts, which can be cultivated naturally. The Agarose chemical structure provides a good support for films. In addition, it is reported that the films, which are made of algae, are transparent, strong, and flexible [2]. In order to enhance the shelf-life and improve the strength of the algae film, cinnamon extract was used to act as a co-primer and anti-microbial in the film. Based on a previous study, anti-microbial polymer film was able to restrain microbial development, hence, broadening the time span of usability of sustenance [6]. Cinnamaldehyde is an organic compound with the formula of C_6_H_5_CH=CHCHO and occurs naturally as a predominant trans (E) isomer, giving cinnamon its flavor and odor [1]. It is a type of flavonoid that is naturally synthesized by the shikimate pathway [2]. This pale yellow, viscous liquid occurs in the bark of cinnamon trees and other species of the genus *Cinnamomum* sp. The essential oil of cinnamon bark contains about 50% cinnamaldehyde [3]. Cinnamaldehyde is also used as a fungicide [8]. Proven effective on over 40 different crops, cinnamaldehyde is typically applied to the root systems of plants [5]. Its low toxicity and well-known properties makes it ideal for agricultural activities. Thus, “cinnamaldehyde” is an effective insecticide, and its scent is also known to repel animals, such as cats and dogs [8]. It has also been tested as a safe and effective insecticide against mosquito larvae [9]. At a concentration of 29 ppm, cinnamaldehyde can kill half of Aedes aegypti mosquito larvae within 24 h [10]. The “trans-cinnamaldehyde” also works as a potent fumigant and practical repellent for adult mosquitos [11]. By adding cinnamon active chemicals into the packaging system [5], the growth rate of microorganisms in food can be inhibited or reduced. Among other antimicrobials, cinnamaldehyde, which is a major component of cinnamon, also possesses antimicrobial activity and has been utilized in the processing of milk, chicken, and meat [5,6]. The objectives of this study were (1) to assess the suitable percentage loading of cinnamon extract with algae film, and (2) to characterize the effect of cinnamon with the biofilm based on the soil bury test, tensile test, FTIR, and SEM.

## 2. Materials and Methods 

Raw algae and cinnamon were purchased from a local store, located in Gombak, Malaysia, while glycerol and acetic acid were then purchased from Sigma Aldrich (Selangor, Malaysia). For the preparation of raw materials of the biofilm, algae were processed though cleaning, drying, and shredding into powder form (ranging from 50 µ–100 µ). The cinnamon oil was collected using the microwave essential oil extraction method. Temperature was set at 60 °C for 6 min as the cinnamon started to steam up and condensation took place for the production of the extraction oil. Next, we let the extraction oil cool down for another 20 min before collecting it. Using a separator, we collected the oil and applied low heat (33–35 °C) to separate the oil and water. After 20 min of heating process, the oil particle moved on top of the water surface. The oil was collected using the pipet and placed in the vial. 

### 2.1. Solution Casting Method

The algae biofilm was prepared using the solution casting method. The ingredients to prepare biofilm with CEO is as follows: 2% algae powder, three different percentages of CEO at 1%, 3%, and 5%; 1.5 mL glycerol solution; 1%, 3%, 5%, 7%, and 9% acetic acid with 0.2% molarity and distilled water were weighed individually using an electronic mass balance. Acetic acid was obtained in liquid form. It was diluted to 0.2% (*w*/*v*) using distilled water. The algae, glycerol, acetic acid, and distilled water were mixed in a beaker, which was then heated up to 90 °C on a hot plate and held at that temperature for 25 min. The stirring speed of the magnetic stirrer was set at a constant speed of 250 rpm, to avoid the formation of bubbles and maintain the homogeneity of the solution. Then, the mixed solution was cooled down to 65 °C for 35 min. During cooling, stirring was continued to prevent the formation of bubbles and solidification of the solution. The second batch of the biofilm was repeated with the addition of the CEO. Pure biofilms (0% CEO) were set up as a control sample. The solution was cast into a square form (18 × 27 cm) of the acrylic plate. Upon casting, the drying process took place in an oven at a temperature of 50 °C for 24 h. The biofilm thickness was measured using an electronic gauge (Digitronic Caliper, Gombak, Selangor, Malaysia), with accuracy ranging between 0.1% and 1% as a function of thickness value (0–100 µm or 0–1000 µm). Seven replicates were made for each type of biofilm formulation.

### 2.2. Soil Burial Test

The compostability of the biofilms and CEO additions were performed according to soil bury test ISO/DIS 17088. The biofilm dimensions of 20 mm × 20 mm were cut and weighed and five replicates were made for each formulation. The bury test area was plotted at a cool and shaded corner of the garden. The soil temperature was based on the normal climate change, which is from 33 to 35 °C, while soil type was black garden soil. Each sample was buried in a convenient depth of 50 mm to allow for aerobic soil bury composting, as the compost has to be turned at regular intervals in this process. The area was plotted with granite or brick to prevent interruption or error during the investigation. Each time the specimen was retrieved from the ground, the plotted area was covered with layers of dried leaves or thin layers of soil to allow air to permeate the hole and accelerate the growth and expansion of fungi or bacteria.

### 2.3. Tensile Test

The Instron tensile test ASTM D882-02 machine (Gombak, Selangor, Malaysia) was used for this test. The load of the machine was set at 5 kN with the speed at 10 mm/min. Seven replicates of strips for each composition were cut at dimensions of 70 mm × 10 mm. The result of the tensile strength and elongation at break were assessed through the graph of the stress-strain curve. 

### 2.4. Fourier Transform Infrared Spectrometer (FTIR)

An FTIR Spectrometer (Perkin Elmer System spectrum 100; PerkinElmer, Gombak, Selangor, Malaysia) is an analytical technique used to identify organic, polymeric, and, in some cases, inorganic materials. The FTIR analysis method uses infrared light to scan test samples and observe chemical properties. The resolution was set up at 4cm^−1^ in a spectral range of 4000 to 600 cm^−1^ and 32 scans per sample. Different peaks (various functional groups of chemical elements) of the IR spectrum were observed along the selected initial angle to the final angle.

### 2.5. Scanning Electron Microscopy (SEM)

The surface morphology of the films was studied using a Scanning Electron Microscope (SEM) JSM 5600 (Gombak, Selangor, Malaysia) with magnifications up to 1000×. Prior to carrying out the observation, the samples were subjected to sputter coating with a layer of carbon using a Polaron SC515 (Gombak, Selangor, Malaysia). This procedure was performed to ensure the sample morphology could be clearly observed under SEM and to prevent any electrostatic charging during observation.

## 3. Results and Discussions

### 3.1. Biofilm Thickness 

In a polymer film packaging application, thickness is a crucial aspect, which requires specific attention from the material design. The thickness of the biofilm will highly influence other important properties, such as the strength, elasticity, and moisture content. Researchers [5] found that the main purpose of effective biofilm for food packaging is to secure the food from food pathogens, thus extending the shelf-life of the food, and will ensure the quality of the food and its nutrients to be intact. The general thickness of biofilms for packaging is ±0.3 mm [12]. Table 1 shows the thickness of the algae-based film by varying algae sample. 

The second batch formulation was focused on biofilm with the addition of several percentages of CEO. Based on this observation, it could be highlighted the importance of using CEO compared to the cinnamon powder. By using CEO, it was easier to control the thickness of the biofilm, as the resulting film thickness was not significantly different compared to the thickness of the control biofilm (0%, as depicted in Table 1). This is because the CEO used in the solution form mixed well within the blends. In comparison, using the cinnamon powder, the course cinnamon particle will not dissolve in the biofilm solution during processing, hence affecting the thickness of the bioplastic. In industry, this parameter is important for food packaging. This trend was similar to the previous study [13], where whey and pectin protein powder were incorporated into cinnamon and researchers had difficulty to control the thickness of the biofilm.

### 3.2. Algae-Based Biofilm with Acetic Acid

#### 3.2.1. Tensile Data

(A) Tensile strength, (B) modulus elasticity and (C) elongation at break of algae-based biofilms were affected by the different percentages of acetic acid, as demonstrated in Figure 1. The first attempt to produce algae-based biofilm failed because the sample was too fragile, as the algae is rich with starch content and becomes hydrophilic in nature. The glycerol was chosen to alter the delicateness of the biofilm. Once again, the biofilm produced was low in strength, as the particles of glycerol may have leaked during processing and the biofilm produced was found to tear easily. Next, the flexibility of the biofilm was improved with the help of the acetic acid content, to aid the glycerol to be fully efficient in the algae-based biofilm. Acetic acid is a weak acid that has one carboxylic acid group and is usually use in food additive [5]. Although there has been no specific research on the usage of acetic acid as crosslinking agents, it has reported that the presence of acetic acid increased the interfacial interaction in the properties of coconut shells filled with low a polyethylene composite [14]. Researchers [15] also indicated that the combination of acetic acid with CO_2_ packaging can extend the shelf life from 12 to 20 days for chicken retail cuts without negatively affecting the quality and sensory properties of the broiler meat. The addition of acetic acid into the blend may help the infusion of glycerol into the algae molecular structure by accelerating the disintegration and suspension of algae sediment. Previous researchers also stated similar strength results in a PVA and chitosan blend with glycerol and acetic acid [16]. However, from the analysis done, the addition of 1%, 3%, 5%, and 7% acetic acid in the biofilm decreases the (A) tensile strength to half compared to the control biofilm, probably due to the different molecular structures of the acetic acid, even though it comes from the same carboxylic acid family [17]. Figure 1 also illustrated a reduction in (B) modulus of algae-based biofilms, similar to the tensile strength, with an increase in the concentration of acetic acid. Based on the figure, the lowest elastic property was recorded for 7% acetic acid, further than this percentage will continuously drop the modulus strength of the biofilm. This finding was similar to previous attempts using glycerol to increase the percentage of citric acid by up to 15% [3]. On the other hand, the (C) elongation at break (Eb) results were vice versa to the tensile and modulus strength results. Based on the Eb graph, it was found that the biofilms were capable of resisting changes in shape without crack formation with the addition of the acetic acid. Figure 1 demonstrated that the (C) elongation at break gradually increased with the increasing percentage of acetic acid by up to 7%. The highest elongation at break was measured with 5% acetic acid at 27.34% of Eb, while the lowest Eb was shown by 0% acetic acid at 20.14%. Physically, the biofilm with the addition of acetic acid was better in flexibility, less fragile, and good in modulus. The biofilm obtained was transparent and could not easily tear off when folded. However, the addition of acetic acid neither improves the tensile strength nor the elongation at break of the biofilm.

#### 3.2.2. Soil Burial Test 

The algae used in this research to form a biofilm is a green algae species known as Neochloris Oleoabundans or Ettlia Oleoabundans [18]. These unicellular green algae are freshwater based and rich in starch content [18,19]. Starch is hydrophilic in nature and easy to degrade due to microbial and moisture contact. The degradation of algae-based bioplastic film via starch by micro-organisms in the soil produced carbon dioxide, biomass formed by extraction of algae carbon, and soluble CEO compound. From the soil bury test analysis, gradual biodegradation was observed in the biofilm surface degradation as shown in Figure 2. Figure 2 demonstrates the physical appearance of the algae biofilm after soil burial test for 28 weeks. The samples for soil bury test were exposed to the actual weather. Under rainy conditions, excess water permeated through the soil and diffused into the biofilm samples causing swelling and softening of the biofilm. Based on the physical observation, the biofilm started to deteriorate after 14 weeks and onwards, most likely due to hydrophilic nature of the algae. At the end of the 28 week period, the sample could barely retain the shape and began to wrinkle and tear apart. From the pictures, the sample showed high number of pores, and the number of pores continues to spread and increase in size as the length of soil bury test was prolonged. This confirmed that the biofilm sample had undergone biodegradation phases. 

### 3.3. Algae-Based Biofilm with CEO 

#### 3.3.1. Tensile Data 

Algae-based biofilm with 3% acetic acid was found to yield good (A) tensile strength and (B) modulus properties, and, therefore, was selected to be used with CEO. This percentage was selected to be used for further investigation alongside the addition of different ranges of CEO (phase two). The percentages of CEO tested were 1%, 3%, 5%, 7%, and 9%. The control sample is label as 0 in the tabulated figures. Figure 3 demonstrates the (A) tensile strength of algae film, which increases with increasing percentages of CEO. The control sample without acetic acid possessed the least tensile strength. Meanwhile, the maximum tensile strength was achieved with 5% CEO due to the good intermolecular interaction between algae and starch and cinnamon molecules. This finding was also supported by a previous study [20], which also recorded a similar pattern where tensile strength increased with the addition of cinnamon bark oil into the alginate film. Based on Figure 3, the (B) modulus elasticity of the algae-based biofilm was found to display the same trend as the tensile strength results. The addition of 5% CEO was found to increase the stiffness of the algae-based biofilm up to 0.323 GPa, compared to algae-based biofilm with 1%, 3%, 7%, and 9% of CEO, which recorded lower modulus strength. However, in this study, the 5% CEO loading did affect the (C) elongation at break compared to the other percentages. Figure 3 indicated that the 5% CEO loading has low elongation at break compared to the 3% and 7% of CEO loading. Based on the current findings, 5% CEO with 3% acetic acid yielded a good and accepted elongation at break of the algae-based biofilm. This was made possible with the right amount of acetic acid in strengthening and adhering to the intermolecular bonds between the algae and cinnamon molecules. Hence, the addition of acetic acid into algae-based biofilm clearly indicated that the acetic acid molecules affect the adjacent molecules by increasing the distance and reducing the internal force, resulting in a more flexible film. The interference with adjacent molecules affects the intermolecular and intramolecular linkage of the polymer, thus strengthening the structure of the algae-based biofilm [21]. 

#### 3.3.2. Soil Burial Test 

Figure 4 depicted the physical appearance and Figure 5 shows the SEM analysis of biofilm with 5% CEO after soil burial at the 7th, 14th, 21th, and 28th weeks, respectively. The analysis is similar to control algae-based biofilm in Figure 2, it was noticed that the color of the biofilms with CEO turned darker and the darkening of the biofilms is a sign of biodegradation [22]. Changes in the appearance of the biofilms are explainable through the high-moisture absorption property and low intensity of cinnamaldehyde in the CEO percentages [23]. Based on the physical appearance of the biofilm in Figure 4, the sample in this research would have behaved similarly to the findings by Zhang et al. [20], where the alginate films incorporated with cinnamon bark oil showed less biodegradation potential compared to the alginate film without cinnamon bark oil. Therefore, it can be postulated that the addition of 3% acetic acid into the recipe assists in reducing the decomposition rate of the biofilm with CEO, compared to the sample of 5% CEO without acetic acid content. From the SEM analysis in Figure 5, the agglomeration of biofilm became more obvious as the biofilm began to swell, which in turn caused slow degradation. The addition of 5% of CEO into the biofilm exhibited physical changes. Besides which, different volumes of CEO used in this study resulted in varying biodegradation rates and behaviors. The antimicrobial (cinnamaldehyde functional group) and repellent properties of cinnamon may also decelerate the degradation rate of the film. These findings are in accordance with the previous research [23], where the higher the cinnamon percentage, the slower the composability rate. The film with 5% CEO demonstrated lesser pore percentages. A higher amount of CEO tends to reduce the degradation rate because of the hydrophobicity of the acetic acid, and the strong aroma of the cinnamon itself may repel insects and micro-organisms from attacking the biofilm [24].

### 3.4. Fourier Transform Infra-Red (FTIR) Spectroscopy Data

Figure 6A illustrated the FTIR spectra of biofilm with 5% CEO content and B biofilm with 3% of acetic acid, respectively, both displaying individual peaks within the range of 4000–500 cm^−1^. Peak A presented a broad absorption band at about 3310 cm^−1^, which represents the hydroxyl (OH) group. The peak at 2924 cm^−1^ was recognized due to the C–H stretching of methane. Besides this, the peak at 1606 cm^−1^ was formed due to the stretching vibration of the conjugated peptide bond formation by amine (NH_2_) and acetone groups in the algae. The peak at 1441 cm^−1^ was due to an ester sulfate group. The characteristic peaks at 1013 cm^−1^ and 931 cm^−1^ indicated C–O stretching groups of 3,6-anhydrogalactose [25]. In addition, based on the FT-IR spectrum of CEO in A, the absorption band or frequency ranged from 3500 cm^−1^ to 3200 cm^−1^ broad, exhibiting the presence of O–H stretch. The specific absorbance band at 1635 cm^−1^ revealed the stretching vibration of the C=O bond for cinnamaldehyde [25]. Due to the influence of conjugation and an aromatic ring, the peak is wider than usual for aldehyde compounds. A strong absorption band between 900 cm^−1^ and 690 cm^−1^ indicated the presence of aromatic C=C bonds [26]. Cinnamaldehyde is the main active component in cinnamon, which can be used as a natural antimicrobial in food preservation to retard or inhibit the bacterial growth of pathogenic and spoilage bacteria, which in turn extends the shelf life of the food products [20]. Since there was no peak observed at 1700–1720 cm^−1^ in Figure 6B, which shows that there was no crosslinking between acetic acid and the algae-based blends due to the absence of chemical reaction. The sighting of a peak at wavenumber ranges between 1700–1720 cm^−1^ indicates the presence of cellulose-fatty acids ν(C=O), a stretching vibration of the esters. The slope was transmitted obviously in (A), however, slowly lowering down with the addition of acetic acid as showed in sample (B). The combination of CEO and acetic acid was significantly reduced the cellulose fatty acid presence in the algae as shown in sample (C) which is possibly occurs due to the formation of a physical reaction between CEO, acetic acid and algae [26]. Figure 6C represents the FT-IR spectrum of 3% acetic acid blends with 5% CEO biofilms. A peak at 1716 cm^−1^ was observed, indicating an association with C=O, which is attributed to the carboxyl and ester carbonyl bands. This confirmed the existence of acetic acid in the specimen. However, the peak at 3328 cm^−1^ became less intense when cinnamon was added into the formulation. 

## 4. Conclusions

The tensile test of biofilm demonstrated good enhancement upon the incorporation of 5% CEO. The biofilm achieved tensile strength at 4.8 MPa and elongation of 15%. Based on the SEM morphology, higher amounts of CEO in the presence of acidic acid leads to a reduction in the degradation rate of the biofilm. The biofilm demonstrated a continuous phase and exhibited a characteristic band at 1716 cm^−1^ in the FTIR analysis. Hence, in conclusion, 5% CEO and 3% acetic acid are the suitable blend that could tremendously affect the tensile behavior and the biodegradation rate of the biofilm.

## Figures and Tables

**Figure 1 polymers-11-00004-f001:**
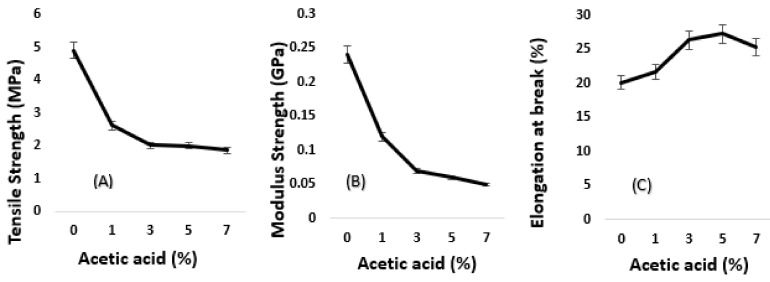
Effect of acetic acid on (**A**) tensile strength; (**B**) modulus strength; and (**C**) elongation at break of algae-based biofilm.

**Figure 2 polymers-11-00004-f002:**
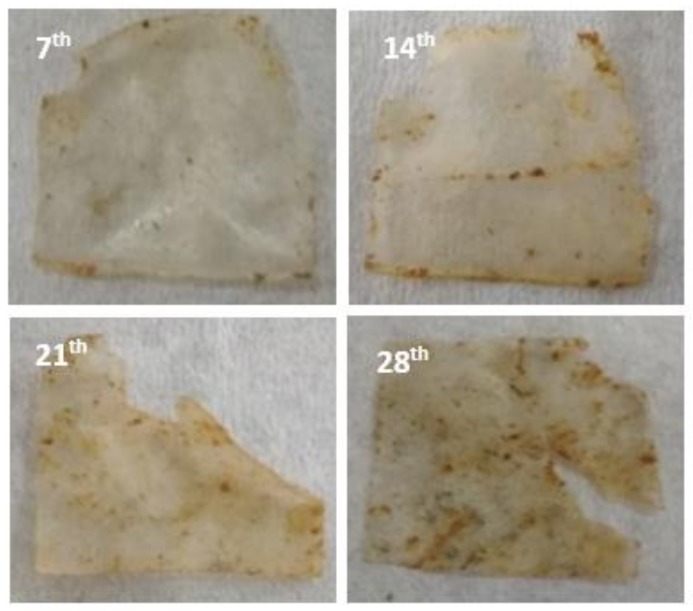
Physical appearance of algae film after soil burial at the **7th**, **14th**, **21th**, and **28th** weeks.

**Figure 3 polymers-11-00004-f003:**
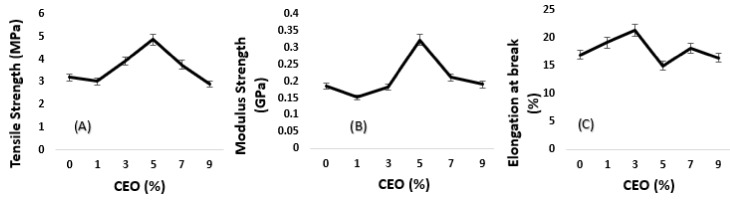
Effect of CEO on (**A**) tensile strength; (**B**) modulus strength; and (**C**) elongation at break of algae-based film.

**Figure 4 polymers-11-00004-f004:**
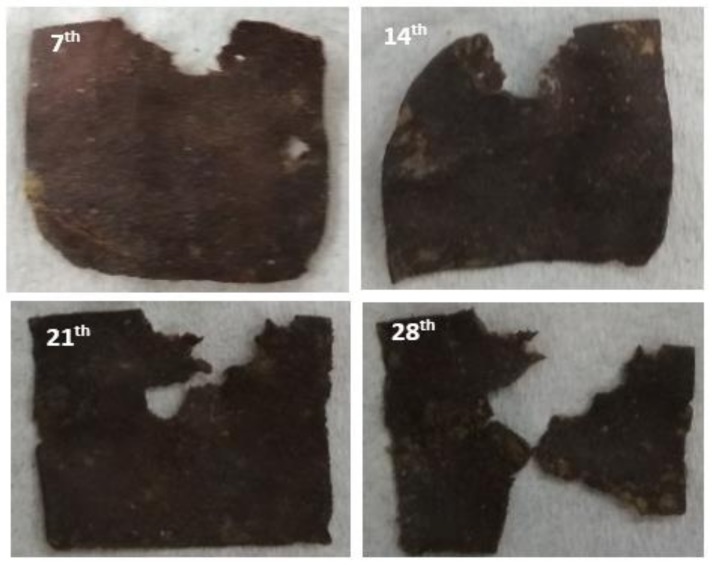
Physical appearance of biofilm 5% of CEO at the **7th**, **14th**, **21th**, and **28th** weeks.

**Figure 5 polymers-11-00004-f005:**
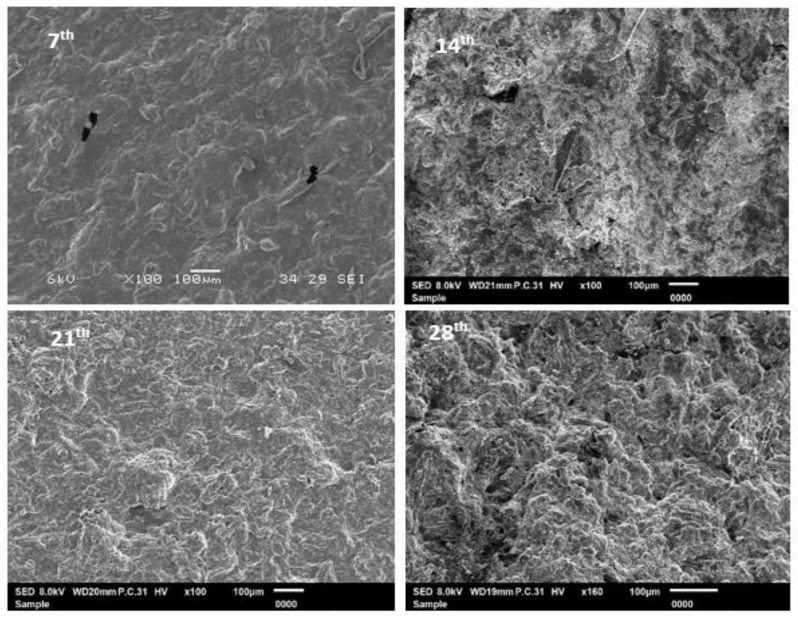
SEM analysis of algae film with 5% CEO after soil burial at the **7th**, **14th**, **21th**, and **28th** weeks.

**Figure 6 polymers-11-00004-f006:**
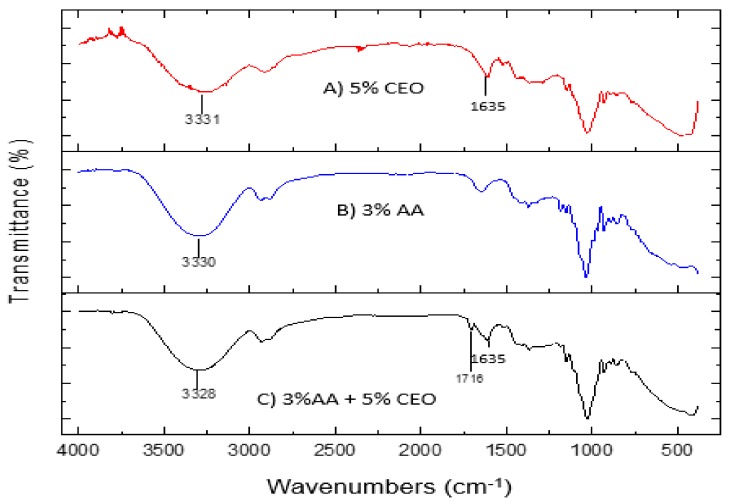
FTIR spectra of (**A**) Biofilm with 5% CEO; (**B**) biofilm with 3% acetic acid; and (**C**) biofilm with 5% CEO and 3% acetic acid.

**Table 1 polymers-11-00004-t001:** Thickness of algae-based film with and without CEO.

Sample (% Algae)	Biofilm without CEO Thickness (mm)	Biofilm with CEO Thickness (mm)
0	0.2 ± 0.01	0.2 ± 0.01
1	0.2 ± 0.01	0.2 ± 0.01
2	0.2 ± 0.01	-
3	0.2 ± 0.01	0.2 ± 0.02
4	0.2 ± 0.02	-
5	0.2 ± 0.01	0.2 ± 0.03

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
