# Peer review of "Effect of Cinnamon Extraction Oil (CEO) for Algae Biofilm Shelf-Life Prolongation"

_polymers, 2018, doi:10.3390/polym11010004_

Round 1

Reviewer 1 Report

It is opinion of the reviewer that his paper before acceptance needs several corrections. My individual comments are listed below.

The authors did not investigate antimicrobial assays in this research. Therefore, I suggest to remove a term of “antimicrobial” from the title as well as from Keywords.

L. 74 - Cinnamon is a spice not a chemical compound.

L. 74 Cinnamaldehyde is not a flavonoid.

L. 81 – It should be “cinnamaldehyde”.

L. 85 – “trans” should be in italic.

L. 93 – Algae should be characterized from the biological point of view.

Equipment used for extraction should described.

L. 96/97 – Wrong style “extracted using …extraction method”.

L. 104/105 – Composition is not clear!

L. 133 – It should be FTIR spectrometer.

L. 134/135 – This sentence should be rephrased.

L. 137 – What does it mean “were observed”?

A section “Statistical analysis” should be completed.

Table 1 & 2 should be merged.

L 176 – First sentence is repeated. Remove it.

Table 3 and 4 are rather figures!

The quality of Figure 3 is very low and needs to be much better.

Author Response

Point 1: The authors did not investigate antimicrobial assays in this research. Therefore, I suggest to remove a term of “antimicrobial” from the title as well as from Keywords.

Response 1: Accept and removed.

Point 2: L. 74 - Cinnamon is a spice not a chemical compound.

Response 2: accepted and amended.

Point 3: L. 74 Cinnamaldehyde is not a flavonoid.

Response 3: accepted and amended.

Point 4: L. 81 – It should be “cinnamaldehyde”.

Response 4: accepted and amended.

Point 5: L. 85 – “trans” should be in italic.

Response 5: Amended.

Point 6: L. 93 – Algae should be characterized from the biological point of view.

Equipment used for extraction should described.

Response 6: accepted and amended.

Point 7: L. 96/97 – Wrong style “extracted using …extraction method”.

Response 7: accepted and amended.

Point 8: L. 104/105 – Composition is not clear!

Response 8: accepted and amended.

Point 9: L. 133 – It should be FTIR spectrometer.

Response 9: accepted and amended.

Point 10: L. 134/135 – This sentence should be rephrased.

Response 10: Rephrased

Point 11: L. 137 – What does it mean “were observed”?

Response 11: Rephrased

Point 12: A section “Statistical analysis” should be completed.

Table 1 & 2 should be merged.

Response 12: Merged

Point 13: L 176 – First sentence is repeated. Remove it.

Table 3 and 4 are rather figures!

Response 13: Removed

Point 14: The quality of Figure 3 is very low and needs to be much better.

Response 14:  Changed to high resolution picture.

Reviewer 2 Report

The addition of CEO is capable of decelerating the degradation and affecting the strength property of the biofilm. In this work, the authors demonstrated that the prolongation shelf-life rate of biofilm with good tensile properties are achievable with the addition of 5% of CEO. Overall, the concept of this paper is clearly presented. I would like to recommend that this manuscript could be published in Polymers after major revision.

Here, I added a few questions and remarks:

1. There are some typos need to be corrected. For example, the first sentence in the abstract, the word “live-span” should be “life-span”.

2. In Table 3, Table 4 and Table 5, the resolution of photos and SEM images is too low. Please provide those images with high quality.

3. In FTIR analyses, the authors described “lower O-H bonds in the algae causes the polymer to become more hydrophobic”. However, with FTIR analyses, the description is not proper. FTIR should not use for quantitation of functional groups in the samples. The authors can measure contact angles of samples to demonstrate the increase of hydrophobicity.

Author Response

Point 1: There are some typos need to be corrected. For example, the first sentence in the abstract, the word “live-span” should be “life-span”.

Response 1: Accept and amended.

Point 2: In Table 3, Table 4 and Table 5, the resolution of photos and SEM images is too low. Please provide those images with high quality.

Response 2: accepted and amended.

Point 3:  In FTIR analyses, the authors described “lower O-H bonds in the algae causes the polymer to become more hydrophobic”. However, with FTIR analyses, the description is not proper. FTIR should not use for quantitation of functional groups in the samples. The authors can measure contact angles of samples to demonstrate the increase of hydrophobicity.

Response 3: accepted and amended.

Reviewer 3 Report

1. In Figure 1, the authors showed effect of acetic acid on tensile strengh of biofilm increased with increasing percentage of acetic acid by up to 0.5%. So what about higher acetic acid up to 1% ? Is 1% better or worse than 0.5% ?

2. In Figure 2. Are these results derived from algae-based biofilm solely with increasing precentage of cinnamon powder or blend of cinnamon with acetic acid ? Because it is mentioned that "0.3% CEO with 0.3% acetic acid yielded a good and accepted elongation at break of the 242 algae-based biofilm." from the result. However, Figure 2 is not clearly clarified the above conclusion. Besides, Is the phase of cinnamon extraction oil (CEO) kind of powder ? Why in Figure 2, different range of cinnamon powder rather than CEO was added ?

3. In Figure 3 FTIR spectroscopy, why 0.1% acetic acid rather than 0.3% acetic acid was used in the blend with 5% CEO ? Which is not consistent with the final conclusion that 5% CEO and 0.3% acetic 338 acid are the suitable blend.

4. As presented, the wrinting is no acceptable for publish. There are problems with sentence structure, verb tense, and clause construction. Some sentences contain grammatical and/or spelling mistakes.

Author Response

Point 1: In Figure 1, the authors showed effect of acetic acid on tensile strength of biofilm increased with increasing percentage of acetic acid by up to 0.5%. So, what about higher acetic acid up to 1%? Is 1% better or worse than 0.5%?

Response 1: Amended, based analysis done, it worsens.

Point 2: In Figure 2. Are these results derived from algae-based biofilm solely with increasing percentage of cinnamon powder or blend of cinnamon with acetic acid? Because it is mentioned that "0.3% CEO with 0.3% acetic acid yielded a good and accepted elongation at break of the 242 algae-based biofilms." from the result. However, Figure 2 is not clearly clarified the above conclusion. Besides, Is the phase of cinnamon extraction oil (CEO) kind of powder? Why in Figure 2, different range of cinnamon powder rather than CEO was added?

Response 2: accepted and amended. It is Cinnamon not powder, it is amended.

Point 3: In Figure 3 FTIR spectroscopy, why 0.1% acetic acid rather than 0.3% acetic acid was used in the blend with 5% CEO? Which is not consistent with the final conclusion that 5% CEO and 0.3% acetic 338 acid are the suitable blend.

Response 3: accepted and amended. Changed to 5% CEO and 3% AA to make consistency with the conclusion.

Point 4: As presented, the wrinting is no acceptable for publish. There are problems with sentence structure, verb tense, and clause construction. Some sentences contain grammatical and/or spelling mistakes.

Response 4: accepted and amended.

Round 2

Reviewer 1 Report

The authors corrected this paper properly taken under considerations all my comments. Therefore I can accept it now.

Reviewer 2 Report

The authors have addressed all my concerns. I would like to recommend to accept this article as its current form.

Reviewer 3 Report

The manuscript has been well revised and I recommend to accept it in the present form.